# GCAL: Adapting Graph Models to Evolving Domain Shifts

**Ziyue Qiao** [* 1]  **Qianyi Cai** [* 2]  **Hao Dong** [3]  **Jiawei Gu** [1]  **Pengyang Wang** [4]  **Meng Xiao** [3]  **Xiao Luo** [5]  **Hui Xiong** [2 6]

## Abstract

This paper addresses the challenge of graph domain adaptation on evolving, multiple out-of-distribution (OOD) graphs. Conventional graph domain adaptation methods are confined to single-step adaptation, making them ineffective in handling continuous domain shifts and prone to catastrophic forgetting. This paper introduces the **G**raph **C**ontinual **A**daptive **L**earning (**GCAL**) method, designed to enhance model sustainability and adaptability across various graph domains. GCAL employs a bilevel optimization strategy. The "adapt" phase uses an information maximization approach to fine-tune the model with new graph domains while re-adapting past memories to mitigate forgetting. Concurrently, the "generate memory" phase, guided by a theoretical lower bound derived from information bottleneck theory, involves a variational memory graph generation module to condense original graphs into memories. Extensive experimental evaluations demonstrate that GCAL substantially outperforms existing methods in terms of adaptability and knowledge retention. The code of GCAL is available at https://github.com/joe817/GCAL.

## 1. Introduction

Graphs are ubiquitously present in the real world, serving as fundamental structures for representing complex systems in a multitude of domains. Graph models, leveraging these in-

---
[*]Equal contribution  [1]School of Computing and Information Technology, Great Bay University [2]Thrust of Artificial Intelligence, The Hong Kong University of Science and Technology (Guangzhou) [3]Computer Network Information Center, University of the Chinese Academy of Sciences [4]University of Macau [5]Department of Computer Science, University of California, Los Angeles [6]Department of Computer Science and Engineering, The Hong Kong University of Science and Technology Hong Kong SAR, China. Correspondence to: Hui Xiong <xionghui@ust.hk>, Xiao Luo <xiaoluo@cs.ucla.edu>, Meng Xiao <shaow@cnic.cn>.

*Proceedings of the 42nd International Conference on Machine Learning*, Vancouver, Canada. PMLR 267, 2025. Copyright 2025 by the author(s).

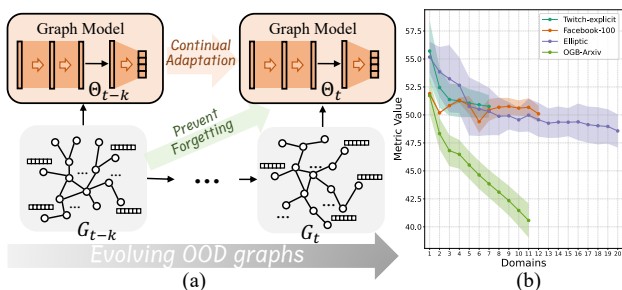

*Figure 1.* (a) The challenge of continual adaptation of graph models on evolving OOD graph sequences. (b) Empirical evaluations of the SOTA graph adaptation method across four OOD graph datasets in a continual adaptation setting.

tricate connections, have been pivotal in advancing data mining and knowledge discovery. Classic graph models such as Graph Convolutional Networks (GCNs) (Kipf & Welling, 2016) and Graph Attention Networks (GATs) (Veličković et al., 2017) have been successfully applied in numerous applications ranging from social network analysis (Dong et al., 2023; Qiao et al., 2022; Sun et al., 2024) to bioinformatics (Huang et al., 2024; Wang et al., 2024c) and recommendation systems (He et al., 2020; Wu et al., 2024b).

Despite the successes, the sustainability of graph models in handling ever-increasing volumes of graph data presents unique challenges, particularly in scenarios involving new, unseen graphs. Such OOD scenarios commonly arise when a model trained on one set of graph data is applied to a different and novel set. This discrepancy underscores a pivotal issue in graph machine learning: *Domain Adaptation in Graph Models*. The objective is to enhance the model's inference ability to generalize across different but related graph distributions without substantial retraining.

However, the current body of research often limits its focus to single-step adaptations employing techniques like Maximum Mean Discrepancy (MMD) (Dziugaite et al., 2015) and adversarial learning (Dan et al., 2024; Qiao et al., 2023; Zhang et al., 2018). While useful, these techniques fall short when the model is subjected to continual domain shifts over time or domains: *As graph datasets grow and evolve, models encounter new domains that necessitate ongoing adaptation but lose their ability to adapt to previous graphs, resulting*

*in catastrophic forgetting.* The problem is illustrated in Figure 1 (a), and as the empirical results depicted in Figure 1 (b), the state-of-the-art (SOTA) graph domain adaptation method EERM (Wu et al., 2022b) experiences a continuous and serious decline in performance across four evolving OOD graph datasets.

Continual learning (Wang et al., 2024a; 2022; Zhang et al., 2022a;b; 2023b), or lifelong learning, offers a promising solution to mitigate catastrophic forgetting. A widely adopted strategy within this paradigm is the replay mechanisms, where the model periodically revisits selected or generated old data to reinforce past knowledge. This practice helps in maintaining a balance between training cost and knowledge retention. However, existing approaches typically rely on labeled data to select and replay memories, presenting a substantial barrier in many applications where such labels are either unavailable or prohibitively expensive to obtain. This introduces a significant research gap: *the development of continual adaptive methods for unsupervised memory generation and replay in graph models.* Such methods would need to autonomously identify critical features and structural patterns within graphs that are essential for the model's long-term adaptability and robustness.

In this paper, we introduce a novel method named Graph Continual Adaptive Learning (GCAL), specifically designed to tackle the challenges of catastrophic forgetting in graph models' continual adaptation. Our approach employs an "adapt and generate memory" bilevel optimization strategy, activated each time new graph data is introduced. For "adapt," we utilize an information maximization approach to adapt the model to new domain graphs, simultaneously re-adapting the previous memory graphs to prevent forgetting. For "generate memory," we theoretically derive a lower bound for preserving informative and generalized memory graphs from the current graph, leveraging the principles of the information bottleneck. The main contributions of this research are outlined as follows:

- We introduce the GCAL framework to effectively manage catastrophic forgetting and enhance the sustainable reuse of graph models during their continual adaptation across evolving OOD graph data.

- We derive a theoretical lower bound that ensures the preservation of informative and generalized memory graphs. Based on this foundation, we design a memory graph generator equipped with three tailored losses to effectively guide the memory graph learning process.

- We conduct extensive experiments on various graph datasets, demonstrating that GCAL significantly outperforms state-of-the-art across domain shifts.

## 2. Preliminary

We present the formulation for the continual adaptive learning on graphs. Given a graph model $f(\Theta_0) : G_s \rightarrow \mathcal{Y}$ pre-trained on one or multiple source graphs for a specific classification task, where $\Theta_0$ represents the pre-trained parameters, $G_s$ is the source graph and $\mathcal{Y} = \{y_1, y_2, ..., y_C\}$ is the set of $C$ classes. The sequence of $m$ target domain graphs is defined as $\{G_1, G_2, ..., G_m\}$, where each graph $G_t = \{A_t, X_t\}$ belong to the $t$-th domain. $A_t \in \mathbb{R}^{N_t \times N_t}$ is the adjacency matrix and $N_t$ are the numbers of nodes. $X_t \in \mathbb{R}^{N_t \times d}$ is the attribute matrices and $d$ are the dimensions of node attributes. In this scenario, the target graphs arrive one by one sequentially, and each target graph may exhibit a different distribution from previous ones due to changes in the underlying data over region and time, i.e., $p(G_i) \neq p(G_j), \forall i \neq j$, where $p(\cdot)$ is the data distribution. We aim at sustainable reusing of the graph model for the continual adaptation and inference on multiple out-of-distribution target domains within the same task in an online fashion. As the target domain graph $G_t$ arrives, we feed the model on $G_t$ to adapt the parameters $\Theta_{t-1} \rightarrow \Theta_t$ and make the prediction accordingly. There are two purposes in the process of test-time training: (1) **Adapting**: we aim to ensure that the model adapts effectively to new target domain graphs as they arise; (2) **Avoid Forgetting**: we aim to retain the model's performance on previously encountered target graphs after each adaptation, all without the need for complete retraining.

## 3. Methodology

As shown in Figure 2, our method first utilizes an information maximization approach to adapt the current model $\Theta_{t-1}$ to the newly arrived graph $G_t$ while simultaneously conducting memory replay on the previous memory graphs to avoid forgetting, which will be introduced in Sec. 3.1. Then the updated model parameter $\Theta_t$ is used to learn the small memory graph $\widehat{G}_t$ for $G_t$, which will be introduced in Sec. 3.2. To generate the memory graph, we develop a variational memory graph generation module comprising a variational GNN, a trainable selector, and a novel graph structure learning and reparameterization technique. To optimize the memory graph, we follow the lower bound to introduce three learning objectives. The memory graph learning loss uses a graph condensation technology to learn task-related memory graphs. The regularization losses are proposed to ensure the stability and informativeness of the memory graph. The generation loss enhances the relevance of the memory graph to the original graph.

### 3.1. Adaptation with Memory Replay

In the $t$-th adaptation step, the goal is to refine the current model $f(\Theta_{t-1})$ to make the prediction to enhance its pre-

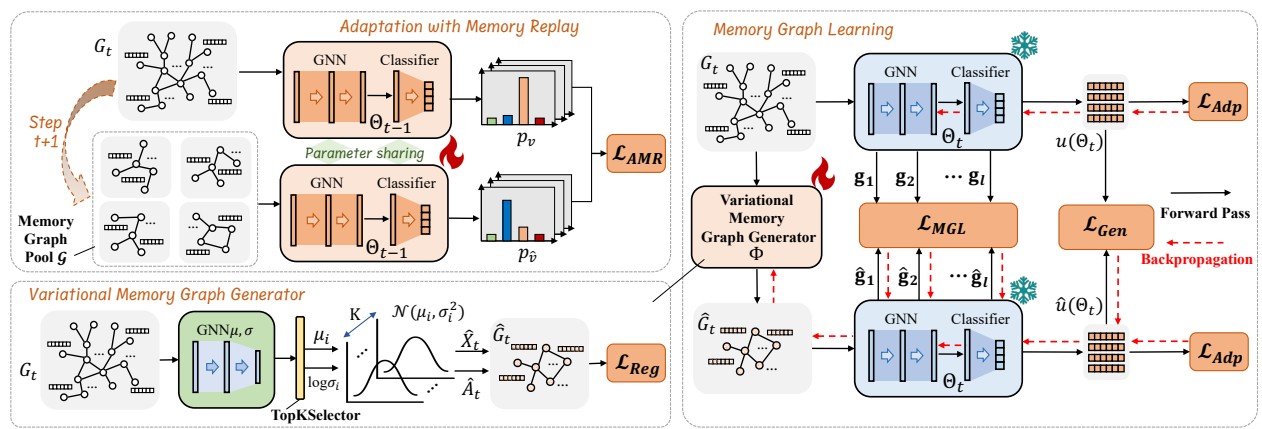

*Figure 2.* The GCAL framework involves several steps: Starting with the graph model $f(\Theta_{t-1})$, the current graph $G_t$, and the accumulated memory graph pool $\mathcal{G} = \{\widehat{G}_i\}_{i=1}^{t-1}$, GCAL first applies the Adaptation with Memory Replay method using loss $\mathcal{L}_{AMR}$ for model adaptation. Next, a Variational Memory Generator creates a new memory graph $\widehat{G}_t$ for $G_t$, which is refined using the memory graph learning loss $\mathcal{L}_{MGL}$, the regularization losses $\mathcal{L}_{Reg}$ to ensure stability and informativeness, and the generation loss $\mathcal{L}_{Gen}$ to enhance the memory graph's relevance to $G_t$. Finally, $\widehat{G}_t$ is added to $\mathcal{G}$ for future adaptation.

dictive accuracy on target domain graphs. Since there is no label on the target graphs, the model adaptation is primarily conducted self-supervised. We adopt the Information Maximization ([Liang et al., 2020](#)) leveraged on the output probability of the model. This approach rests on the fundamental premise: *A model that effectively discriminates target data will exhibit high inferential confidence, characterized by output probabilities that closely resemble a one-hot vector*. With the output probability $p_v$ of each node $v$ encoded from pre-trained model $f(\Theta_{t-1})$, the objective to minimize risk (see Eq. 1) on is as follows:

$$\mathcal{L}_{Adp}(G; \Theta_{t-1}) = -\mathbb{E}_{v \sim \mathcal{V}}\left[\sum_{k=1}^{C} p_{v,k} \log(p_{v,k})\right] + \sum_{k=1}^{C} \widehat{p}_k \log \widehat{p}_k, \quad (1)$$

where $p_{i,k}$ denotes the $k$-th element of $p_i$. The expected probability $\widehat{p}_k$ is calculated as $\widehat{p}_k = \mathbb{E}_{v \in \mathcal{V}}[p_{v,k}]$. The second term introduces a diversity regularization designed to enhance the variety of output probabilities. This regularization helps prevent the issue where a few high integrity scores might dominate during training, potentially causing all unlabeled nodes to converge towards the same pseudo-label, resulting in overfitting.

To prevent the model from forgetting previously learned graphs while adapting to new graphs, known as memory replay in continual learning, we apply the information maximization loss not only to the new graph but also to all previous graphs. For efficient replay, we use a graph memory pool, denoted as $\mathcal{G} = \{\widehat{G}_1, \widehat{G}_2, \ldots, \widehat{G}_{t-1}\}$. This pool contains a sequence of smaller synthetic graphs, each representing a previously encountered graph. We then perform adaptation using memory replay, formulated as follows:

$$\mathcal{L}_{AMR} = \mathcal{L}_{Adp}(G_t; \Theta_{t-1}) + \sum_{i=1}^{t-1} \mathcal{L}_{Adp}(\widehat{G}_i; \Theta_{t-1}). \quad (2)$$

By combining the adaptation loss on the new graph $G_t$ with the adaptation losses on the graphs in the memory pool $\mathcal{G}$, the model $f(\Theta_{t-1})$ is continually refined to adapt to new graphs while reinforcing performance on historical graphs.

## 3.2. Variational Memory Graph Generation

In our model, meanwhile adapting to the new graph $G_t$ with memory-aware replay, our goal is also to learn the memory $\widehat{G}_t$ corresponding to $G_t$, forming a series of memories $\mathcal{G}_t$ that can be replayed when the next adaptation task arrives to prevent forgetting. We consider three factors: *(1) the memory size should be significantly smaller than the original graph, (2) the memory should be informative, retaining as much important information from the source graph as possible, and (3) the memory should be generalizable, capable of being stored across diverse graph distributions.*

### 3.2.1. DERIVING INFORMATION BOTTLENECK ON MEMORY GRAPHS

To achieve the above motivations, we proposed a variational information bottleneck based memory graph generation method. Graph information bottleneck ([Sun et al., 2022](#); [Wu et al., 2020b](#)) usually aims to maximize the below:

$$\widehat{G}_t = \arg\max_{\widehat{G}_t}\left[I(\widehat{G}_t; \widehat{Y}_t) - \beta I(\widehat{G}_t; G_t)\right], \quad (3)$$

where $I(\cdot; \cdot)$ denotes the mutual information and $\widehat{Y}_t$ represents the training signals associated with nodes in $\widehat{G}_t$. The first term, $I(\widehat{G}_t; \widehat{Y}_t)$, aims to preserve task-related information within the memory, while the second term, $I(\widehat{G}_t; G_t)$, focuses on compressing information from the original graph into the smaller memory graph, effectively filtering out irrel-

evant information. $\beta$ acts as a trade-off parameter, balancing the compression of input data with the preservation of task-relevant information.

Despite the goal of $\widehat{G}_t$ to preserve information from the graph, directly generating it from $G_t$ is challenging. Instead, we generate $\widehat{G}_t$ from the original graph through a variational latent representation $Z_t$, expressed as $P_g(\widehat{G}_t|G_t) = P_g(\widehat{G}_t|Z_t, G_t)P_g(Z_t|G_t)$ where $g(\Phi)$ is the generator with parameter $\Phi$. Utilizing this chain rule, we can reformulate the second term in Eq.3 as: $I(\widehat{G}_t; G_t) = I(\widehat{G}_t; G_t, Z_t) - I(\widehat{G}_t; Z_t|G_t)$. Consequently, the optimization objective for generating $\widehat{G}_t$ can be reformulated as follows:

$$\mathcal{L}(\Phi) = \max_{\Phi} \left[ I(\widehat{G}_t; \widehat{Y}_t) - \beta I(\widehat{G}_t; G_t, Z_t) + \beta I(\widehat{G}_t; Z_t|G_t) \right].$$

(4)

To optimize this objective in a parameterized manner, we derive a lower bound in the following Theorem:

**Theorem 3.1.** *Let $\widehat{G}_t$ be a generated graph conditioned on the latent representation $Z_t$ of the original graph $G_t$. Suppose $Q(\widehat{G}_t)$ is a variational approximation of the true posterior $P(\widehat{G}_t)$, Then, the following lower bound on the optimization objective for $\Phi$ holds:*

$$\mathcal{L}(\Phi) \geq \mathbb{E}[\log P_f(\widehat{Y}_t|\widehat{G}_t)] - \beta \mathbb{E}[KL(P_g(\widehat{G}_t|G_t, Z_t) \parallel Q(\widehat{G}_t))]$$
$$+ \beta \mathbb{E}[\log(P_g(\widehat{G}_t|G_t, Z_t))].$$

(5)

*Here, $KL(\cdot \parallel \cdot)$ indicates the Kullback-Leibler divergence. $P_f$ is considered as the classifier $f(\Theta_t)$.*

The proof can be found in Appendix A. Thus, the memory graph learning objective can be maximizing the above lower bound. In the following sections, we first introduce the variational memory generator. Then, we introduce the optimization objectives for each item in Equation 5 in detail.

### 3.2.2. VARIATIONAL MEMORY GRAPH GENERATOR

In Theorem 3.1, we define the memory graph generator as $g(\Phi) : P(G_t) \rightarrow P(\widehat{G}_t)$. We first employ a GNN architecture to process the input graph $G_t$, transforming it into latent distributions:

$$[\mu; \log \sigma] = \text{TopKSelector}(\text{GNN}_{\mu,\sigma}(A_t, X_t)),$$
$$\text{TopKSelector}(X) = \underset{X}{\text{argsort}} \left( \text{Sigmoid} \left( \frac{X\mathbf{p}}{\|\mathbf{p}\|} \right) \right) [: K],$$

(6)

where $\mu \in \mathbb{R}^{K*h}$ and $\log \sigma \in \mathbb{R}^{K*h}$ represent the mean and variance components for each node, respectively. $K \ll N_t$ is the number of nodes in the generated graph, and $h$ is the hidden dimension. $\text{GNN}_{\mu,\sigma}(\cdot)$ is parameterized to output a vector of dimensions $2 * h$, divided into mean and variance components. To manage graph dimensionality and emphasize significant distributions, a top-k selector layer $\text{TopKSelector}(\cdot)$ reduces the number of distributions,

where $\mathbf{p} \in \mathbb{R}^h$ is its trainable parameters. We compute the logarithm of the standard deviation (i.e., $\log \sigma$) rather than directly calculating $\sigma$, which smoothly scales the deviation, enhancing numerical stability and interpretability.

We first generate the latent variable of each node of the memory graph from the distribution via the reparameterization trick:

$$\widehat{z}_i \sim \mathcal{N}\left(\widehat{z}_i|\mu_i, \sigma_i^2\right) = \mu_i + \sigma_i^2 \odot \varepsilon,$$

(7)

where $i = 1, ..., K$ and $\varepsilon \in \mathcal{N}(0, I)$ is a random variable drawn from a standard normal distribution. This reparameterization ensures that the sampling process remains differentiable, allowing the gradients to be backpropagated through the sampling step during training.

Then, we assume $\widehat{z}_i$ as the node features in the memory graph and obtain the feature matrix via $\widehat{X}_t = \text{id}([\widehat{z}_i]_{i=1}^K)$ where $\text{id}(\cdot)$ is the identity function. We further generate the edges of the memory graph from $\widehat{z}_i$. We assume that each edge follows an independent Bernoulli distribution, with each edge characterized by a binary random variable $a_{i,j} \sim Bernoulli(w_{i,j})$ for each edge. We use $\widehat{z}_i$ to generate the learnable Bernoulli weights for each edge. Given that $a_{i,j}$ is non-differentiable to $w_{i,j}$, we approximate it as a continuous variable within the interval [0,1]. To facilitate gradient-based optimization, the Gumbel-Max reparameterization trick, as detailed by (Maddison et al., 2017), is employed to update the edges as follows:

$$w_{i,j} = \frac{(\text{MLP}([\widehat{z}_i; \widehat{z}_j]) + \text{MLP}([\widehat{z}_j; \widehat{z}_i]))}{2},$$
$$a_{i,j} = \text{Sigmoid} \left( (w_{i,j} + \log \frac{\delta}{1-\delta})/\tau \right),$$

(8)

where $\delta \sim \text{Uniform}(0, 1)$ and $\tau$ represents the temperature hyperparameter. As $\tau$ approaches 0, $a_i$ becomes increasingly binary. The reparameterization enables a well-defined gradient, $\frac{\partial a_{i,j}}{\partial w_{i,j}}$, allowing for effective training of $w_{i,j}$. Consequently, $a_{i,j}$ can be obtained through the training process and used as the edge weight in constructing the adjacency matrix $\widehat{A}_t$ for the memory graph.

In this way, by combining the above modules together, we obtain the variational memory graph generator $g(\Phi)$ and the memory graph is obtained by $\widehat{G}_t = g(G_t, \Phi) = \{\widehat{A}_t, \widehat{X}_t\}$. Leveraging the variational approach not only aids in the generation of nodes and edges but also helps in managing and optimizing the underlying distributions of these elements efficiently.

### 3.2.3. MEMORY GRAPH LEARNING VIA CONDENSATION LOSS

The first term in Eq.5, $\mathbb{E}[\log P_f(\widehat{Y}_t|\widehat{G}_t)]$, involves maximizing the expected log-likelihood of the predicted outcomes given the generated memory graphs $\widehat{G}_t$. This can

be reframed as minimizing the condensation loss (Jin et al., 2021), a novel objective that enhances the fidelity and relevance of the generated graphs to downstream tasks:

$$\min_{\Phi} \mathcal{L}(f(\widehat{G}_t; \Theta_t), \widehat{Y}_t), \quad \widehat{G}_t \in \mathbb{G}_t$$
$$\text{s.t.} \quad \Theta_t = \arg\min_{\Theta} \mathcal{L}(f(G_t; \Theta_{t-1}), Y_t), \tag{9}$$

where $\mathcal{L}$ is the task-related loss on the graphs and corresponding training signals. As $\widehat{Y}_t$ and $Y_t$ are not directly observable, we instead use the adaptation loss in Eq.1 with the soft pseudo-labels as training signals. In previous approaches, the gradient matching scheme was often employed to minimize this loss. This method aligns the gradients of the model trained on the generated memory graph $\widehat{G}_t$ with the gradients from the true graph $G_t$ with respect to the network parameters $\Theta_t$. By doing so, it ensures that the model's behavior on the generated data closely mirrors its behavior on actual data, facilitating better generalization:

$$\mathcal{L}_{MGL} = \min_{\Phi} D\left(\frac{\partial \mathcal{L}_{Adp}(\widehat{G}_t; f(\Theta_t))}{\partial \Theta_t}, \frac{\partial \mathcal{L}_{Adp}(G_t; f(\Theta_t))}{\partial \Theta_t}\right), \tag{10}$$

where $D(\cdot, \cdot)$ is the sum of the distance between gradients at each layer. Given two gradients $\widehat{\mathbf{g}} \in \mathbb{R}^{d_1 \times d_2}$ and $\mathbf{g} \in \mathbb{R}^{d_1 \times d_2}$ at a specific layer, the distance between them is defined as:

$$D(\widehat{\mathbf{g}}, \mathbf{g}) = \sum_{i=1}^{d_2} \left(1 - \frac{\widehat{\mathbf{g}}_i \cdot \mathbf{g}_i}{\|\widehat{\mathbf{g}}_i\| \|\mathbf{g}_i\|}\right), \tag{11}$$

where $\widehat{\mathbf{g}}_i, \mathbf{g}_i$ are the $i$-th column vectors of the gradient matrices. With this optimization, we are able to achieve task-related memory graph learning through the efficient gradient-matching strategy.

### 3.2.4. REGULARIZATION LOSS

The second term in Eq.5, $\mathbb{E}[\text{KL}(P_g(\widehat{G}_t|G_t, Z_t) \| Q(\widehat{G}_t))]$, corresponds to the KL divergence measure the learned distribution $P_g$ of the predicted graph $\widehat{G}_t$, given the actual graph $G_t$ and latent variables $Z_t$, deviates from a simpler prior $Q(\widehat{G}_t)$. We refine the prior distribution $Q(\widehat{G}_t)$ by distinguishing between the components of node features and edges, i.e., $Q(\widehat{G}_t) = Q(\widehat{A}_t, \widehat{X}_t) = Q(\widehat{A}_t) \cdot Q(\widehat{X}_t)$. Thus, the optimization objective of the term can be rewritten as minimizing the following:

$$\min_{\Phi} \mathbb{E}[\text{KL}(P_g(\widehat{A}_t|G_t, Z_t) \| Q(\widehat{A}_t))] $$
$$+ \mathbb{E}[\text{KL}(P_g(\widehat{X}_t|G_t, Z_t) \| Q(\widehat{X}_t))]. \tag{12}$$

For the first term, as outlined in Section 3.2.2, we define the edges to follow an independent Bernoulli distribution. Accordingly, we specify $Q(\widehat{A}_t)$ such that each edge $a_{i,j}$ adheres to a Bernoulli distribution $Bernoulli(q)$, where $q$ is

the predefined probability parameter. Additionally, consistent with prevailing approaches in the literature, we define $Q(\widehat{X}_t)$ for node features as a Normal Gaussian distribution $\mathcal{N}(0, I)$, where $I$ represents the identity matrix in $\mathbb{R}^{h \times h}$. Then, the overall regularization loss can be defined as follows:

$$\mathcal{L}_{Reg} = \min_{\Phi} \frac{1}{2} \sum_{i=1}^{K} \sum_{j=1}^{h} \left(\mu_{i,j}^2 + \sigma_{i,j}^2 - \log(\sigma_{i,j}^2) - 1\right) + $$
$$\sum_{i,j=1}^{K} \left(w_{i,j} \log \frac{w_{i,j}}{q} + (1 - w_{i,j}) \log \frac{1 - w_{i,j}}{1 - q}\right). \tag{13}$$

This approach acts as the regularization term that ensures that the variability introduced during the generation of new memory graphs is effectively controlled, leading to more stable adaptations over successive domains.

### 3.2.5. GENERATION LOSS

For the last terms Eq.5, the objective becomes minimizing $-\mathbb{E}[\log(P_g(\widehat{G}_t|G_t, Z_t))]$, the likelihood of generating the graph $\widehat{G}_t$. In conventional methods, the original graph usually serves as a reference for optimizing the generated graph, typically employing a reconstruction-based discrepancy loss to quantify the differences between the generated and the original graphs. However, our approach deviates from traditional methods due to the reduced size of the generated graph, challenging direct structural and feature-based comparisons. To address this, we adopt a distribution-based discrepancy measure, specifically designed to assess differences in the aggregate properties of the graphs:

$$\mathcal{L}_{Gen} = \min_{\Phi} \text{Dis}(\widehat{G}_t, G_t) = \left\|\sum_{i=1}^{K} \widehat{u}_i(\Theta) - \sum_{i=1}^{N_t} u_i(\Theta)\right\|_2, \tag{14}$$

where $\text{Dis}(\cdot)$ means the discrepancy between the memory graph and the original graph, $\widehat{u}_i(\Theta), u_i(\Theta) \in \mathbb{R}^{h'}$ are the hidden representations of nodes encoded from the model $f(\Theta)$ before the final classification head. Note that the node representations are different from those in Eq.7, which are encoded by the variational generator. $\|\cdot\|_2$ denotes the L2 normalization distance. We did not use MMD or adversarial alignment methods but a more concise measure to minimize the distribution discrepancy because the detailed distribution learning has already been effectively optimized in the former modules, and additionally, our method is more efficient and robust.

### 3.3. Optimization

Combining the loss of adaptation with memory replay and the three losses in memory generation, we can establish the overall learning objective as a bi-level optimization frame-

work. When the $t$-th graph $G_t$ arrives, the learning contains two stages, the inner loop aims to adapt the model $f(\Theta_{t-1})$ on $G_t$ while the outer loop aims to learn a memory graph $\widehat{G}_t$ based on the adapted model $f(\Theta_t)$:

$$\min_{\widehat{G}_t, \Phi} \mathcal{L}_{MGL}(G_t, \Theta_t; \Phi) + \lambda_1 \mathcal{L}_{Reg}(G_t; \Phi) + \lambda_2 \mathcal{L}_{Gen}(G_t, \Theta_t; \Phi),$$

$$\text{s.t.} \quad \Theta_t = \arg\min_{\Theta} \mathcal{L}_{AMR}(G_t, \{\widehat{G}_i\}_{i=1}^{t-1}; \Theta_{t-1}),$$

$$(15)$$

where $\lambda_1, \lambda_2$ are the loss weights. For the generator, for each timestep, a new generator is utilized to create the memory graph, and only the memory graphs are preserved in the memory buffer. We also use an exponential moving average (EMA) strategy (Wang et al., 2022) to update the mode parameters to smooth the parameter updates.

# 4. Experiments

## 4.1. Experimental Setup

**Datasets** Our paper involves two primary categories of graph datasets, differentiated by regional and temporal shifts. For regional shifts, Facebook-100(Traud et al., 2012) and Twitch-Explicit(Rozemberczki et al., 2021) datasets consist of multiple social networks from different regions. For temporal shifts, OGB-Arxiv(Hu et al., 2020) is a paper citation network dataset, and Elliptic(Pareja et al., 2020) is a Bitcoin transactions network dataset, both of which include graphs from different time steps. In these datasets, each graph is treated as a separate domain. We select certain domains to pre-train a graph model and then adapt it continuously using the remaining graphs.

**Baselines**. We evaluate the performance of our continual adaptive learning framework against a diverse set of baseline methods. Test employs a pretrained graph model to perform direct inference on the target dataset without any adaptation, serving as the lower bound. The category "No Rehearsal Based Test-Time Adaptation" comprises one-step test-time adaptation methods, including DANN (Ganin et al., 2016), Tent (Wang et al., 2021), BN Stats Adapt (Li et al., 2016), EERM (Wu et al., 2022b), and GTRANS (Jin et al., 2022). The category "Continual Test Time Adaptation" refers to continual test-time training methods, including CoTTA (Wang et al., 2022) and EATA (Niu et al., 2022). For baselines not originally designed for graphs, their architectures have been adapted to GCNs to ensure consistency in evaluation.

**Evaluation Metrics**. We report the performance matrix $M^{\text{result}} \in \mathbb{R}^{T \times T}$, which is a lower triangular matrix where $M_{i,j}^{\text{result}}$ (for $i \geq j$) represents the performance on the domain $j$ after training on the domain $i$. Specifically, similar to (Jin et al., 2022; Wu et al., 2022b), for the Twitch-Explicit and Facebook-100 datasets, the results are measured using ROC-AUC and Accuracy, respectively. For the Ellip-

*Table 1.* The statistics of datasets with distribution shifts, where # denotes "the number of".

| Category | Datasets | #Nodes | #Edges | #Domains |
|---|---|---|---|---|
| Regional Shifts | Twitch-explicit | 1,912 - 9,498 | 31,299 - 153,138 | 7 |
| | Facebook-100 | 769 - 41,554 | 33,312 - 2,724,458 | 12 |
| Temporal Shifts | Elliptic | 1,089 - 7,880 | 1,168 - 9,164 | 41 |
| | OGB-Arxiv | 4,427 - 39,711 | 1,225 - 38,735 | 11 |

tic dataset, the metric used is the F1 Score, while for the OGB-Arxiv dataset, Accuracy is used. To compute a single numeric value upon completing all domains, we calculate the Average Performance (AP) as $\frac{1}{T} \sum_{i=1}^{T} M_{T,i}^{\text{result}}$, primarily assessing adaptation ability, and the Average Forgetting (AF) as $\frac{1}{T-1} \sum_{i=1}^{T-1} (M_{T,i}^{\text{result}} - M_{i,i}^{\text{result}})$, primarily evaluating the ability to avoid forgetting. Each experiment is repeated five times, with results reported as the mean and standard deviation. Detailed introduction for datasets, baselines, and experimental settings is in Appendix B.

## 4.2. Experimental Results

### 4.2.1. OVERALL PERFORMANCE COMPARISON.(RQ1)

This experiment aims to answer: *How does GCAL perform in the unsupervised continual adaptation setting across evolving graph data?* We compare GCAL with various baselines divided by different domain adaptation strategies and report the experimental results in Table 2. Note that Test does not involve modifying the model; EERM and GTrans train a new parameter at each graph, inapplicable to previous ones. Thus, their AF results are not applicable. Obviously, we observe that GCAL sets a new state-of-the-art, surpassing all baseline methods across all datasets.

Specifically, certain baseline methods, especially traditional domain adaptation methods, demonstrate poor results. This can be attributed to the difficulty of the problem, which involves various timesteps of unsupervised continual adaptation. This setting requires the model to dynamically adapt to new domains while concurrently retaining knowledge of previous ones. Specifically, when a model fails to preserve knowledge from the current domain, accumulating errors may disrupt performance in future domains and even cause the model to degrade over time. Our approach mitigates this challenge and collectively enhances the sustainable reuse of graph models and effectively alleviates catastrophic forgetting across evolving graph data via the Adaptation with Memory Replay framework. Among the advanced baselines, the most comparable method to GCAL is CoTTA. Our method outperforms CoTTA primarily by integrating the variational memory graph generation module. Instead of solely using EMA for model updates, we employ the Variational Memory Generator to generate previous graphs that enhance knowledge transfer. This strategy not only lever-

*Table 2.* Data performance comparison across four datasets, evaluated against eight baseline methods and GCAL. Test represents the lower bound, while Full indicates the upper bound. N/A: Not Applicable.

| Methods | Twitch-explicit | | Facebook-100 | | Elliptic | | OGB-Arxiv | |
|---|---|---|---|---|---|---|---|---|
| | AP-AUC/%↑ | AF/%↑ | AP-ACC/%↑ | AF/%↑ | AP-F1/%↑ | AF/%↑ | AP-ACC/%↑ | AF/%↑ |
| Test | 53.88±0.00 | N/A | 50.55±0.00 | N/A | 53.97±0.00 | N/A | 42.43±0.00 | N/A |
| DANN | 51.50±0.39 | 0.19±0.55 | 50.63±0.64 | 0.38±1.13 | 53.84±0.56 | -0.95±0.54 | 42.62±0.23 | -1.28±0.16 |
| Norm | 52.65±0.00 | -2.30±0.00 | 47.62±0.00 | 0.53±0.00 | 54.67±0.00 | 0.02±0.00 | 40.21±0.00 | 0.11±0.00 |
| TENT | 52.84±0.98 | -1.83±1.05 | 46.63±0.00 | 0.61±0.00 | 46.54±0.00 | 0.55±0.00 | 36.72±0.37 | -0.02±0.18 |
| EERM | 52.08±0.00 | N/A | 49.70±0.03 | N/A | 46.51±0.00 | N/A | 36.26±0.00 | N/A |
| GTrans | 53.55±0.29 | N/A | OOM | N/A | 54.25±0.02 | N/A | 39.69±0.04 | N/A |
| CoTTA | 53.94±0.36 | 0.34±0.41 | 50.12±0.16 | 0.59±0.12 | 54.08±0.05 | -1.92±0.06 | 40.28±0.01 | -1.96±0.01 |
| EATA | 53.56±0.10 | -0.72±0.07 | 49.02±3.16 | -0.57±0.03 | 50.48±0.07 | -0.76±0.06 | 40.91±0.70 | -1.35±0.59 |
| GCAL | **55.65±0.09** | **0.42±0.13** | **52.72±0.36** | **0.72±0.19** | **56.57±0.14** | **0.88±0.13** | **45.22±0.17** | **0.76±0.05** |

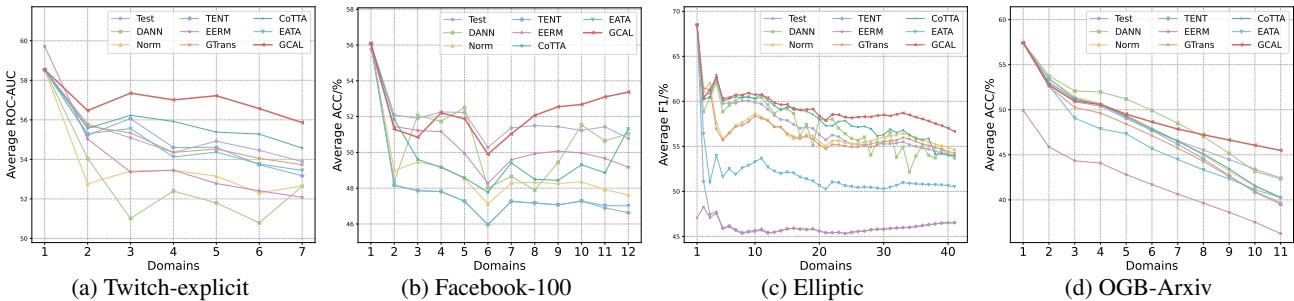

| (a) Twitch-explicit | (b) Facebook-100 | (c) Elliptic | (d) OGB-Arxiv |
|---|---|---|---|

*Figure 3.* Dynamics of the average performance during continual adaptation on evolving OOD graphs.

ages past insights for improved adaptation but also enriches the training process, boosting the model's adaptability and robustness across evolving domains. DANN achieves relatively high performance among the baselines as it uses the source graph to guide the adaptation, while others only use the target graph data. Notably, our method surpasses DANN in four datasets, particularly in AP. This improvement is attributed to GCAL's enhanced ability to adapt effectively to new target domains, leverage knowledge from past experiences for better adaptation, and maintain performance on previously encountered domains, all without the need for complete retraining. This novel approach significantly bolsters the model's performance and adaptability.

### 4.2.2. IN-DEPTH ANALYSIS OF CONTINUOUS PERFORMANCE.(RQ2)

This experiment aims to answer: *How does GCAL's fine-grained performance evolve after continuously learning each domain?* To present a more fine-grained demonstration of the model's performance in continual adaptive learning on graphs, we analyzed the average performance across all previously encountered domains each time a new domain was learned. The comparative results of Test, DANN, GCAL, and the top-performing baseline are depicted in Figure 3. The curve represents the model's performance after $t$ in terms of AP on all previous $t$ tasks. Also, we visualize the

accuracy matrices of GCAL and CoTTA on the Twitch and Elliptic datasets. The results are presented in Figure 4. In these matrices, each row represents the performance across all domains upon learning a new one, while each column captures the evolving performance of a specific domain as all domains are learned sequentially. In the visual representation, darker shades signify better performance, while lighter hues indicate inferior outcomes.

From the results, we observed that as the number of domains increases, the learning objectives grow increasingly complex, resulting in a reduction in performance across all examined methods. That is because as domains accumulate and the learning objectives become multifaceted, it becomes challenging for models to maintain optimal performance across all domains. Notably, most baselines experienced a substantial decline, with the model collapsing with the arrival of merely a few new domains, demonstrating that catastrophic forgetting occurs almost immediately when the model fails to access previous domain data. This reinforces the need for effective continual learning techniques on the sequential graphs where new domains frequently emerge. While the performance drop was observed across all methods, GCAL demonstrated resilience and outperformed the top-performing baseline CoTTA. Also, GCAL predominantly displays lighter shades across the majority of blocks compared to CoTTA in Figure 4. Moreover, its

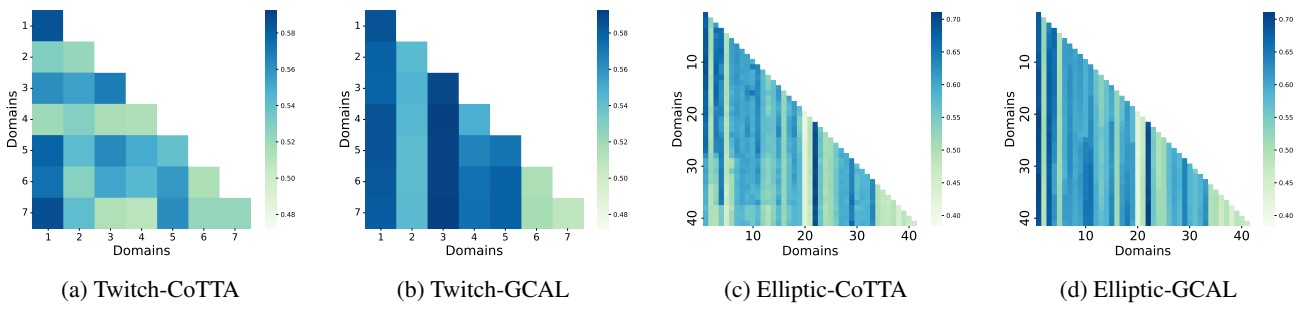

| (a) Twitch-CoTTA | (b) Twitch-GCAL | (c) Elliptic-CoTTA | (d) Elliptic-GCAL |

*Figure 4.* Performance matrices of GCAL and CoTTA in different datasets.

*Table 3.* The results of the ablation study.

| Methods | Twitch-explicit | Facebook-100 | Elliptic | OGB-Arxiv |
|---|---|---|---|---|
| w/o $\mathcal{L}_{Reg}$ & $\mathcal{L}_{Gen}$ | 54.03±2.63 | 52.05±0.31 | 46.53±0.01 | 44.70±0.06 |
| w/o $\mathcal{L}_{Reg}$ | 55.34±0.41 | 52.37±0.56 | 55.23±0.32 | 44.76±0.48 |
| w/o $\mathcal{L}_{Gen}$ | 55.37±0.33 | 52.14±0.32 | 55.64±0.57 | 44.91±0.11 |
| w/o EMA | 54.79±0.04 | 47.66±0.06 | 53.83±0.20 | 43.19±0.08 |
| GCAL | **55.65±0.09** | **52.72±0.36** | **56.57±0.14** | **45.22±0.17** |

competitive performance in specific datasets signifies its robustness and capability. This could be attributed to the "adapt and generate memory" framework, which not only retains critical knowledge from previous tasks but also adapts to new ones.

### 4.2.3. ABLATION STUDIES.(RQ3)

This experiment aims to answer: *Do all the proposed components of GCAL contribute effectively to continual adaptation on graphs?* For that, we design four variant methods for GCAL to verify the EMA in the model parameter updating module, regularization loss, and generation loss in the variational memory graph generation module: w/o $\mathcal{L}_{Reg}$ & $\mathcal{L}_{Gen}$, w/o $\mathcal{L}_{Reg}$, w/o $\mathcal{L}_{Gen}$, and w/o EMA, where "w/o" means "without" the corresponding losses or components in model continual adapting. The results are presented in Table 3. Firstly, we can observe that when each of the losses or components is removed, the model's performance decreases across all datasets; while combining all modules, the method achieves the best results, providing straightforward evidence that all the proposed techniques contribute to our method. For w/o $\mathcal{L}_{Gen}$ the generation loss, the method still achieves remarkable results, primarily due to regularization loss and EMA updates. These components help optimize the latent space representation and ensure smooth model updates, thereby enhancing generalization across diverse data distributions.

### 4.2.4. HYPERPARAMETER EXPERIMENTS.(RQ4)

This experiment aims to answer: *How do synthetic ratio impact the performance of GCAL?* With the Variational Memory Generator, we generate synthesized graphs

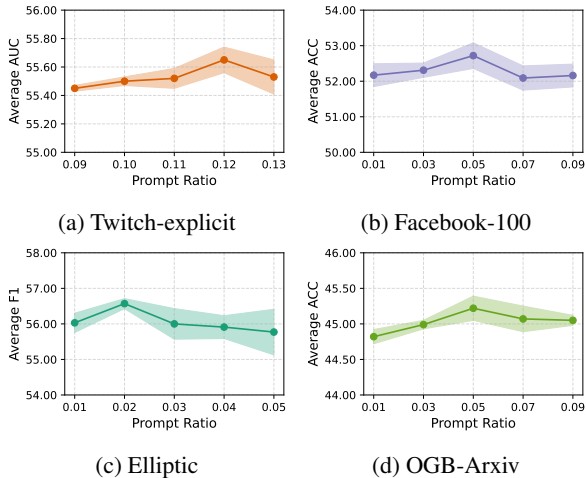

| (a) Twitch-explicit | (b) Facebook-100 |
| (c) Elliptic | (d) OGB-Arxiv |

*Figure 5.* The model performance with different synthetic ratios.

with $K$ nodes for memory replay, where $K$ is much less than the number of nodes in the initial graph. Thus, we set $K$ as $0.01, 0.03, 0.05, 0.07, 0.09$ of the number of nodes for the datasets Facebook-100 and OGB-Arxiv , and as $0.01, 0.02, 0.03, 0.04, 0.05$ for Elliptic, and $0.09, 0.10, 0.11, 0.12, 0.13$ for Twitch, respectively. From the results in Figure 5, we observe that even with a low ratio of synthetic nodes, our model performs well across all datasets. In particular, for Facebook-100 and OGB-Arxiv, which have a larger number of nodes, the model continues to achieve promising results even at lower synthetic node ratios. This demonstrates that our method effectively preserves the information from previous source distributions, as captured by $\mu$ and $\sigma$, highlighting its efficiency and robustness.

## 5. Conclusion

In conclusion, GCAL addresses the critical challenge of unsupervised continual adaptation to out-of-distribution graph sequences. By employing a bilevel optimization strategy, GCAL effectively manages domain shifts and prevents catastrophic forgetting. The method fine-tunes models on new

domain graphs while reinforcing past memories through the adaptation phase, and generates informative and relevant memory graphs guided by a theoretical framework rooted in information bottleneck theory. Experimental results indicate significant improvements over existing methods in terms of adaptability and knowledge retention, enhancing model sustainability and adaptability. A potential limitation of GCAL is its lack of improvement in the graph model architecture itself. This could potentially hinder performance when the base model itself is less capable or inadequate for complex graph data scenarios.

## Impact Statement

This paper presents work whose goal is to advance the field of Machine Learning, with a particular focus on model transfer learning under continual distribution shifts in structured data. There are many potential societal consequences of our work, none of which we feel must be specifically highlighted here. This research has been conducted with a commitment to ethical standards and presents no ethical concerns.

## Acknowledgments

The work is partially supported by the National Natural Science Foundation of China (No. 62406056), the Guangdong Basic and Applied Basic Research Foundation (No. 2024A1515140114, No. 2023B1515120057), the Beijing Natural Science Foundation (No. 4254089), the National Key R&D Program of China (No. 2023YFF0725001), the National Natural Science Foundation of China (No. 92370204), and Guangzhou-HKUST(GZ) Joint Funding Program (No. 2023A03J0008), Education Bureau of Guangzhou Municipality. The computational resources are supported by SongShan Lake HPC Center (SSL-HPC) in Great Bay University.

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

# A. Proof of Theorem 3.1.

In the section of Variational Memory Graph Generation, we give Theorem 1 to define a lower bound of the information bottleneck for generating memory graphs $\widehat{G}_t$ from the original graph $G_t$:

**Theorem A.1.** *Let $\widehat{G}_t$ be a generated graph conditioned on the latent representation $Z_t$ of the original graph $G_t$. Suppose $Q(\widehat{G}_t)$ is a variational approximation of the true posterior $P(\widehat{G}_t)$, Then, the following lower bound on the optimization objective for $\Phi$ holds:*

$$\mathcal{L}(\Phi) \geq \mathbb{E}[\log P_f(\widehat{Y}_t|\widehat{G}_t)] - \beta \mathbb{E}[KL(P_g(\widehat{G}_t|G_t, Z_t) \parallel Q(\widehat{G}_t))] + \beta \mathbb{E}[\log(P_g(\widehat{G}_t|G_t, Z_t))] \tag{16}$$

*Here, $KL(\cdot \parallel \cdot)$ indicates the Kullback-Leibler divergence. $P_f$ is considered as the classifier $f(\Theta_t)$.*

Here we provide the proof of Theorem 3.1:

*Proof.* We start by decomposing the objective function $\mathcal{L}(\Phi)$ in 4 and using variational approximations and known inequalities to make the optimization tractable.

First, the mutual information term $I(\widehat{G}_t; \widehat{Y}_t)$ quantifies the information shared between $\widehat{G}_t$ and $\widehat{Y}_t$, which is defined as:

$$I(\widehat{G}_t; \widehat{Y}_t) = \mathbb{E}_{\widehat{G}_t, \widehat{Y}_t}\left[\log \frac{P(\widehat{Y}_t|\widehat{G}_t)}{P(\widehat{Y}_t)}\right]. \tag{17}$$

Expanding and simplifying this using the definitions of expectation and entropy, we get the following:

$$\begin{aligned} I(\widehat{G}_t; \widehat{Y}_t) &= \mathbb{E}_{\widehat{G}_t, \widehat{Y}_t}[\log P(\widehat{Y}_t|\widehat{G}_t)] - \mathbb{E}_{\widehat{Y}_t}[\log P(\widehat{Y}_t)] \\ &= \mathbb{E}_{\widehat{G}_t, \widehat{Y}_t}[\log P(\widehat{Y}_t|\widehat{G}_t)] + H(\widehat{Y}_t) \\ &\geq \mathbb{E}_{\widehat{G}_t, \widehat{Y}_t}[\log P_f(\widehat{Y}_t|\widehat{G}_t)]. \end{aligned} \tag{18}$$

The inequality holds because $H(\widehat{Y}_t)$, the entropy of $\widehat{Y}_t$, is always non-negative.

Second, the mutual information $I(\widehat{G}_t; G_t, Z_t)$ measures the amount of information gained about $\widehat{G}_t$ by observing $G_t$ and $Z_t$. It is defined as:

$$I(\widehat{G}_t; G_t, Z_t) = \mathbb{E}_{\widehat{G}_t, G_t, Z_t}\left[\log \frac{P(\widehat{G}_t|G_t, Z_t)}{P(\widehat{G}_t)}\right]. \tag{19}$$

We introduce a variational approximation $Q(\widehat{G}_t)$ to the true posterior $P(\widehat{G}_t)$. Then, the negative mutual information is defined as:

$$\begin{aligned} -I(\widehat{G}_t; G_t, Z_t) &= -\mathbb{E}_{\widehat{G}_t, G_t, Z_t}\left[\log \frac{P(\widehat{G}_t|G_t, Z_t)}{Q(\widehat{G}_t)}\right] + \mathrm{KL}(P(\widehat{G}_t) \parallel Q(\widehat{G}_t)) \\ &\geq -\mathbb{E}_{\widehat{G}_t, G_t, Z_t}\left[\log \frac{P(\widehat{G}_t|G_t, Z_t)}{Q(\widehat{G}_t)}\right] \\ &= -\mathbb{E}[\mathrm{KL}(P_g(\widehat{G}_t|G_t, Z_t) \parallel Q(\widehat{G}_t))]. \end{aligned} \tag{20}$$

Third, the conditional mutual information $I(\widehat{G}_t; Z_t|G_t)$ quantifies the additional information about $\widehat{G}_t$ obtained from $Z_t$ given $G_t$. It is defined by the equation:

$$\begin{aligned} I(\widehat{G}_t; Z_t|G_t) &= \mathbb{E}_{\widehat{G}_t, Z_t, G_t}\left[\log \frac{P(\widehat{G}_t, Z_t|G_t)}{P(\widehat{G}_t|G_t)P(Z_t|G_t)}\right] \\ &= \mathbb{E}_{\widehat{G}_t, Z_t, G_t}\left[\log \frac{P(\widehat{G}_t|Z_t, G_t)}{P(\widehat{G}_t|G_t)}\right] \\ &= \mathbb{E}_{\widehat{G}_t, Z_t, G_t}\left[\log P(\widehat{G}_t|Z_t, G_t)\right] + H(\widehat{G}_t|G_t) \\ &\geq \mathbb{E}_{\widehat{G}_t, Z_t, G_t}\left[\log P(\widehat{G}_t|Z_t, G_t)\right]. \end{aligned} \tag{21}$$

The inequality holds because the entropy $H(\widehat{G}_t|G_t)$ is always non-negative.

By combining these results, we can derive the lower bound for the objective $\mathcal{L}(\Phi)$:

$$\mathcal{L}(\Phi) \geq \mathbb{E}[\log P_f(\widehat{Y}_t|\widehat{G}_t)] - \beta\mathbb{E}[\text{KL}(P_g(\widehat{G}_t|G_t, Z_t) \parallel Q(\widehat{G}_t))] + \beta\mathbb{E}[\log(P_g(\widehat{G}_t|G_t, Z_t))] \tag{22}$$

This completes the proof. □

## B. Detialed Experimental Setup

### B.1. Datasets

Our paper involves two primary categories of graph datasets, Regional Shifts and Temporal Shifts, utilizing continual learning principles to effectively manage adaptations across different regions and temporal variations. The datasets used include Facebook-100(Traud et al., 2012), Twitch-Explicit(Rozemberczki et al., 2021), OGB-Arxiv(Hu et al., 2020), and Elliptic(Pareja et al., 2020). The statistical details of the datasets are shown in Table 1.

**Regional Shifts**: The Facebook-100 dataset comprises 100 snapshots of Facebook friendship networks from 2005, each representing users from a specific American university. These networks vary greatly in size, density, and degree distribution. Additionally, the Twitch-Explicit dataset includes seven networks, where nodes are Twitch users and edges denote mutual friendships. Each network originates from one of the following regions: DE, ENGB, ES, FR, PTBR, RU, and TW.

**Temporal Shifts**: The OGB-Arxiv dataset contains 169,343 Arxiv CS papers from 40 subject areas, detailing their citation relationships, and is suitable for analyzing the evolution of scientific collaboration networks. The Elliptic dataset includes 49 sequential graph snapshots of Bitcoin transaction networks, where nodes represent transactions and edges represent payment flows. Around 20% of transactions are labeled as licit or illicit, with the objective of detecting future illegal transactions.

### B.2. Baselines

We evaluate the performance of our continual adaptive learning framework against a diverse set of baseline methods. Test employs a pretrained graph model to perform direct inference on the target dataset without any adaptation, serving as the lower bound. DANN (Ganin et al., 2016), a traditional domain adaptation method, utilizes the source graph and adversarial training to minimize domain discrepancies at each step. The category "No Rehearsal Based Test-Time Adaptation" comprises one-step test-time adaptation methods. Tent (Wang et al., 2021), which minimizes entropy of test samples; BN Stats Adapt (Li et al., 2016), which adjusts network weights and Batch Normalization statistics based on current input data for prediction; EERM (Wu et al., 2022b), tailored for graph datasets, maximizing risk variance to manage domain shifts and out-of-distribution challenges; and GTRANS (Jin et al., 2022), enhancing test-time performance by refining graph structure and node features through a contrastive surrogate loss. The category "Continual Test Time Adaptation" is continual test-time training methods, including CoTTA and EATA. CoTTA (Wang et al., 2022) combines weight-averaged predictions with partial neuron restoration to mitigate error accumulation and catastrophic forgetting. EATA (Niu et al., 2022) enhances adaptation efficiency through entropy minimization and employs a Fisher-based regularizer to maintain performance across domain shifts. For baselines not originally designed for graphs, their architectures have been adapted to GCNs to ensure consistency in evaluation.

### B.3. Experimental Setting

To effectively evaluate our model's adaptability across varying domains and over time, we have introduced a continual adaptive learning framework across evolving graph data. For academic and social networks, initial training is conducted on three university graphs from the Facebook-100 dataset: Amherst41, Caltech36, and Johns Hopkins55. We continually extend the domain adaptation to the remaining eleven university graphs without labels. This data selection sequence requires the model to handle different graph structures from training/validation to testing data(Wu et al., 2022b). For Twitch-explicit, the model is initially trained on the DE network and later tested for adaptability across regional networks, including ENGB, ES, FR, PTBR, RU, and TW, also without labels. In addressing temporal shifts, the OGB-Arxiv dataset is employed, using data prior to 2011 to pretrain the model, while data from 2011 and later is used for continual adaptation, exposing the model to evolving scientific collaborations. For the Elliptic dataset, we utilize snapshots 7 through 9 for initial training, avoiding the first six due to their extreme class imbalance, with the remaining data employed for continual adaptation. This approach

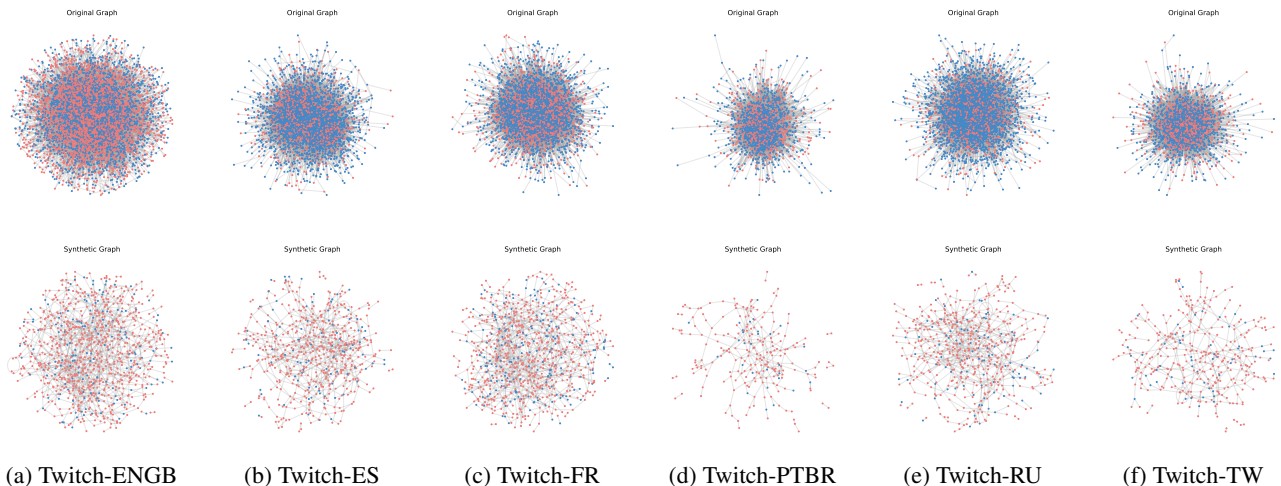

| (a) Twitch-ENGB | (b) Twitch-ES | (c) Twitch-FR | (d) Twitch-PTBR | (e) Twitch-RU | (f) Twitch-TW |

*Figure 6.* Visualized comparison of the original graphs (the first line) and generative graphs (the second line) of GCAL in the Twitch dataset.

reflects the dynamic and challenging nature of financial transactions. Our training strategy begins by pretraining the model on selected graphs from each dataset, then continually adapting to the remaining unlabeled graph datasets in an online manner.

We use a 2-layer GCN as the backbone for three datasets, except for OGB-Arxiv, where we use GraphSAGE (Hamilton et al., 2017). The training, validation, and test rates for pretrain datasets are set at 60%, 20%, and 20% respectively. During the train and adapt periods, the learning rates and weight decays are set as follows: lr = 0.0001 and wd = $5 \times 10^{-4}$ for training, and lr = 0.001 and wd = $5 \times 10^{-4}$ for adaptation. The number of epochs for pre-training is set between 100 and 200, while the adaptation phase involves a relatively smaller number of epochs, ranging from 1 to 10, for these four datasets. The detailed hyper-parameter settings are provided in the accompanying code. For the evaluation metric, we present the accuracy matrix $M_{acc} \in \mathbb{R}^{T \times T}$, which is a lower triangular matrix where $M_{acc,i,j}$ (for $i \geq j$) represents the accuracy on the domain $j$ after training on the domain $i$. Specifically, similar to (Jin et al., 2022; Wu et al., 2022b), for the Twitch-Explicit and Facebook-100 datasets, the results are measured using ROC-AUC and Accuracy, respectively. For the Elliptic dataset, the metric used is the F1 Score, while for the OGB-Arxiv dataset, Accuracy is used. To compute a single numeric value upon completing all domains, we calculate the Average Performance (AP) as $\frac{1}{T} \sum_{i=1}^{T} M_{T,i}^{\mathrm{acc}}$, primarily assessing adaptation ability, and the Average Forgetting (AF) as $\frac{1}{T-1} \sum_{i=1}^{T-1} (M_{T,i}^{\mathrm{acc}} - M_{i,i}^{\mathrm{acc}})$, primarily evaluating the ability to avoid forgetting. Each experiment is repeated five times, with results reported as the mean and standard deviation.

## C. Additional Experiments

### C.1. Visualization of Generated Memory Graph.

This experiment aims to answer: *How do the structure of memory graphs generated by GCAL look like compared with the original ones?* To demonstrate the effectiveness of the memory graphs we generated, we conducted an experiment to visualize the graph structures. The experimental results are shown in Figure 6, where we used the networks python library as a tool on the Twitch dataset to display the generated effects on six continuous domain graphs. The top row shows the original graphs, while the bottom row displays the generated graphs. From this, we can observe that (1) the generated memory graphs significantly reduce the number and density of the graphs, making them more lightweight, and (2) the structure of the generated graphs is not random but shows a coherent structure, verifying the reliability and authenticity of the generated graphs. This visualization confirms the capability of GCAL to produce streamlined yet structurally meaningful graphs.

### C.2. Performance Matrices.

To present a more fine-grained demonstration of the model's performance in continual adaptive learning on graphs, we analyzed the average performance across all previously encountered domains each time a new domain was learned. We have visualized the performance matrix of the Twitch and Elliptic datasets in Figure. 4. Here, we further report the results

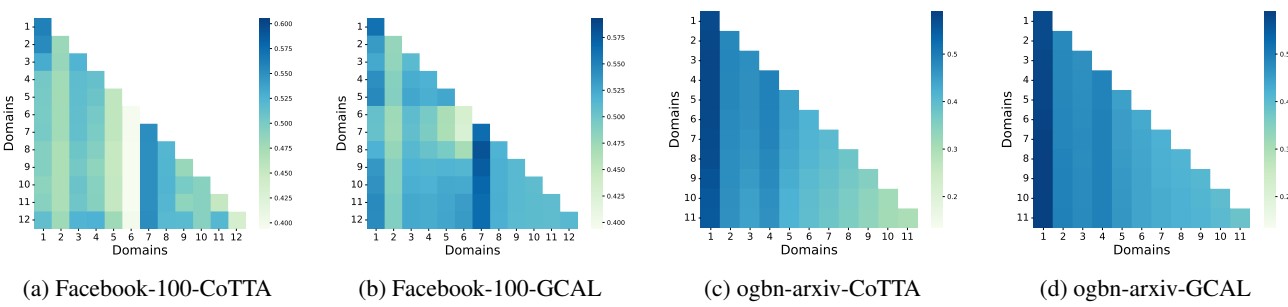

(a) Facebook-100-CoTTA     (b) Facebook-100-GCAL     (c) ogbn-arxiv-CoTTA     (d) ogbn-arxiv-GCAL

*Figure 7.* Performance matrices of GCAL and CoTTA in different datasets.

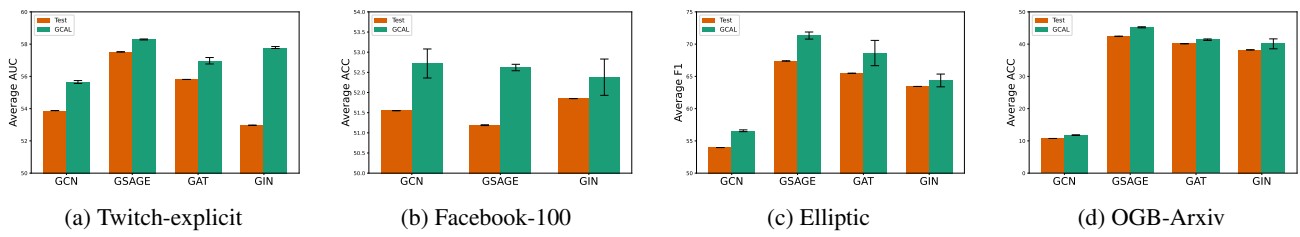

(a) Twitch-explicit     (b) Facebook-100     (c) Elliptic     (d) OGB-Arxiv

*Figure 8.* The model performance with different GNN backbones.

of the Facebook and Ogbn-arxiv datasets in Figure 7. In these matrices, each row represents the performance across all domains upon learning a new one, while each column captures the evolving performance of a specific domain as all domains are learned sequentially. In the visual representation, darker shades signify better performance, while lighter hues indicate inferior outcomes. GCAL predominantly displays lighter shades across the majority of blocks compared to CoTTA in Figure 7, which has similar experimental phenomena as before, further proving the effectiveness of GCAL in the continual graph model adaptation problem.

### C.3. Backbone Analysis.

This experiment aims to answer: *How do different GNN backbones compare within GCAL for continual adaptation?* We select representative GNN models, including GCN, GSAGE, GAT, and GIN. These GNN models serve as the foundational backbones for integrating with GCAL. The results are presented in Figure 8. We compare the results of incorporating these GNN backbones within our framework versus utilizing them individually as standalone models. Using our framework demonstrates remarkable enhancements, showing the effectiveness of our proposed techniques. Lastly, it is important to note that the consistent use of different backbones significantly enhances results, thereby demonstrating the robustness and adaptability of our proposed method across various datasets. This is in contrast to the Test method, which directly infers from subsequent data without continuous domain adaptation. For Facebook-100, the GAT model encounters an Out Of Memory (OOM) issue and is therefore not included in the backbone comparison for this dataset.

## D. Related Work

### D.1. Graph Continual Learning.

Existing graph continual learning (GCL) methodologies are typically divided into three main categories: regularization, parameter isolation, and memory replay approaches. Regularization-based methods primarily aim to preserve parameters crucial to previous tasks, thereby minimizing disruptions (Cai et al., 2022; Liu et al., 2021; Sun et al., 2023a; Xu et al., 2020). Examples include topology-aware weight preserving (TWP) (Liu et al., 2021) and RieGrace (Sun et al., 2023a), which focus on maintaining essential parameters and structural topologies. Parameter isolation techniques allocate distinct parameters for new tasks to maintain those relevant to prior tasks (Niu et al.; Zhang et al., 2023a; 2022a), as seen in HPNs (Zhang et al., 2022a). In contrast, memory replay strategies(Li et al., 2024; Qiao et al., 2025) archive and revisit representative data from past tasks to alleviate the critical issue of catastrophic forgetting, as exemplified by ER-GNN (Zhou & Cao, 2021), SSM (Zhang et al., 2022b), SEM-curvature (Zhang et al., 2023b), PDGNNs (Zhang et al., 2024), and CaT (Liu et al., 2023b).

GCL has garnered increasing interest due to its practical applications, with each approach offering distinct strategies for handling task progression in graph-based models(Wu et al., 2024a). Our approach, which belongs to the memory replay category, uniquely preserves critical topological structures while minimizing memory usage. Notably, while existing GCL methods are confined to supervised learning settings, our work introduces an unsupervised approach to graph continual learning, marking a pioneering step in this direction.

### D.2. Graph Domain Adaptation.

Unlike traditional domain adaptation, which typically assumes a static target domain, Continual Domain Adaptation addresses evolving target data. Traditional methods, including Maximum Mean Discrepancy (MMD) (Dziugaite et al., 2015) and adversarial techniques (Dan et al., 2024; Qiao et al., 2023; Tzeng et al., 2017; Zhang et al., 2018), form the basis of this field. In graph-based domain adaptation, a variety of methods have been proposed (Ding et al., 2018; Jin et al., 2022; Liu et al., 2023a; Ma et al., 2019; Qiao et al., 2024; Wu et al., 2022a; Xiao et al., 2022). For instance, UDA-GCN (Wu et al., 2020a) and AdaGCN (Dai et al., 2022) leverage graph topology to improve adaptability, reducing discrepancies between source and target graphs via local and global consistencies and a graph domain discriminated loss, respectively. Continual Test-Time Adaptation (CTTA), a critical facet of Continual Domain Adaptation, addresses the unique demands of non-static domains. Unlike traditional Test-Time Adaptation (TTA), CTTA incorporates advanced strategies such as bi-average pseudo labels and stochastic weight resets, as implemented in CoTTA (Wang et al., 2022). Innovations like VDP (Gan et al., 2023), with visual domain prompts to counter error accumulation, and RMT (Döbler et al., 2023), which employs symmetric cross-entropy for enhanced robustness, further refine CTTA. Additional strategies, including entropy minimization by Tent and EATA (Niu et al., 2022) and meta-networks in EcoTTA (Song et al., 2023), contribute to improved model normalization and adaptability. Despite these advancements, challenges such as noisy pseudo-labels and calibration issues persist, and a notable gap remains in unsupervised graph continual domain adaptation research.

### D.3. Graph Condensation

Graph condensation(Sun et al.) has become increasingly prominent for its ability to create compact synthetic datasets that closely approximate the performance of full datasets (Hashemi et al., 2024). Classical techniques like GCond (Jin et al., 2021) and MCond (Gao et al., 2024) utilize gradient alignment to synthesize representative samples that maintain the statistical properties of the original data, drawing from principles of traditional sampling (Sener & Savarese, 2017). Among recent innovations, GCDM (Liu et al., 2022), introduce graph-specific distribution alignment to enhance condensation effectiveness. SFGC (Zheng et al., 2024) further refines this by distilling large graphs into structure-free node sets using meta-matching and dynamic feature scoring, resulting in compact, highly generalizable data. Additionally, techniques such as CaT (Liu et al., 2023b) and PUMA (Liu et al., 2023c) extend graph condensation to continual learning, demonstrating these methods' adaptability for dynamic graph-based applications. In parallel, Graph prompt learning based method(Sun et al., 2023b;c; Wang et al., 2024b; Zhao et al., 2024) also condenses knowledge into a compact graph-based prompt, which is typically used to fine-tune pre-trained graph models on specific downstream tasks. However, most of these method rely on supervised signals for knowledge condensation. In contrast, our work addresses a more challenging task by condensing graphs into memory in an unsupervised manner.

