# OpenReview forum: "GCAL: Adapting Graph Models to Evolving Domain Shifts"
_ICML.cc/2025/Conference — ICML 2025 poster_

### Official Review · Reviewer_Ufiz · 2025-03-12

**Overall Recommendation:** 4

**Summary:**

This paper introduces GCAL, a novel framework designed to address the challenge of continual domain adaptation in graph models, particularly in scenarios involving evolving, out-of-distribution graphs. GCAL employs a bilevel optimization strategy: the "adapt" phase fine-tunes the model on new graph domains while mitigating catastrophic forgetting through memory replay, and the "generate memory" phase condenses original graphs into smaller, informative memory graphs using a variational memory generator guided by information bottleneck theory. Extensive experiments on regional and temporal graph datasets demonstrate that GCAL outperforms state-of-the-art methods in adaptability and knowledge retention.

**Claims And Evidence:**

The claims made in the submission are generally well-supported by the theoretical grounding and experimental results.

**Essential References Not Discussed:**

None

**Experimental Designs Or Analyses:**

The experimental designs and analyses in the paper are generally sound and valid. However, the clarifications of why the AF results of some baselines are N/A in Table 2 should be provided.

**Methods And Evaluation Criteria:**

Yes, the proposed methods and evaluation criteria in the paper are well-suited for the problem.

**Other Comments Or Suggestions:**

None.

**Other Strengths And Weaknesses:**

S1. The paper is well-written, with clear problem formulation and illustrations. The motivation is well-justified and compelling.

S2. The paper is grounded in a solid theoretical framework, leveraging information bottleneck theory to derive a lower bound for memory graph generation.

S3: The experiments are extensive, showing the advanced performance of the proposed method compared to baselines.


W1. It appears that the label is not given in the adaptation process, however, the label Y is explicitly referenced in the theoretical analysis. More explanation about how the labels are eliminated in this process should be added.

W2. It is not clear how the variational memory graph generator is superior to the traditional graph generation method.

W3. A deeper analysis of how the weights of losses $L_{Reg}$ and $L_{Gen}$ influence the continual adaptation process would be beneficial.

**Questions For Authors:**

Q1. Why are the AF results of some baseline N/A in Table 2?

**Relation To Broader Scientific Literature:**

This paper is related to graph machine learning, domain adaptation, and continual learning. The paper’s approach makes contributions by addressing the gaps in handling evolving graph domains and mitigating catastrophic forgetting.

**Theoretical Claims:**

The proof of Theorem 3.1 is generally sound and applies information bottleneck theory and variational inference techniques to derive the lower bound.

---

> ### Author Rebuttal · Authors · 2025-03-27
>
> > **W1. It appears that the label is not given in the adaptation process, however, the label Y is explicitly referenced in the theoretical analysis. More explanation about how the labels are eliminated in this process should be added.**
>
> We appreciate the reviewer’s observation. Indeed, in our setting, true labels are not available during the adaptation process, as we focus on unsupervised continual adaptation to out-of-distribution graph domains. Thus, label variable $ \hat{Y}_t$ appears in the theoretical formulation in Equation. 3–5 is addressed via soft pseudo-labels derived from the model‘s predictions.
>
> Specifically, In Section 3.2.1 (Eq. 3–5), we derive a variational lower bound based on the graph information bottleneck:
>
>  $L(\Phi) =  \max_{\Phi}  \left[I(\widehat{G}_t;\widehat{Y}_t) - \beta I(\widehat{G}_t;G_t,Z_t) + \beta I(\widehat{G}_t; Z_t|G_t)\right].$
>
> Here in the first term, $ \hat{Y}_t$ refers to the training signal associated with the memory graph $\widehat{G}_t$, and is used to theoretically quantify task-relevant information retained during memory graph generation. Since true labels are not accessible during adaptation, we set $ \hat{Y}_t$ with soft pseudo-labels generated via self-supervised information maximization in Equation 1. We have clarified this in Section 3.2.3, where the condensation loss is minimized using pseudo-label-based adaptation objectives.
>
> This practice of using pseudo-labels has strong **precedents** in unsupervised domain adaptation and test-time training literature, such as Tent [1] and CoTTA [2]. We follow the same line of reasoning, adapting it to the graph domain.
>
> [1] Wang, Dequan, et al. "Tent: Fully Test-Time Adaptation by Entropy Minimization." ICLR 2021
>
> [2] Wang, Qin, et al. "Continual test-time domain adaptation." CVPR 2022.
>
>
> > **W2. It is not clear how the variational memory graph generator is superior to the traditional graph generation method.**
>
> Thank you for raising this important point. We would like to clarify that the variational memory graph generator in GCAL is specifically designed to address the **unique requirements of continual domain adaptation in graphs**, which traditional graph generation methods do not fully support.
>
> Conventional generators typically focus on reconstructing entire input graphs or producing realistic samples based on learned distributions. However, these methods are not optimized for the goals of continual learning—particularly the need to retain and condense task-relevant knowledge for replay across evolving domains. In contrast, our variational memory graph generator is grounded in the information bottleneck principle, which explicitly balances compression and relevance. It learns a variational latent representation of the input graph and selectively generates a small, condensed graph that captures only the most informative features and structural signals necessary for downstream prediction tasks. It can also generate diversified representations of memory graphs via variational reparameterization for generalized training.
>
> The empirical results in Table 2 and the ablation study in Table 3 demonstrate that memory graphs generated by our variational method enable superior adaptation and retention performance compared to baseline approaches, even with a significantly reduced memory graph.
>
> > **W3. A deeper analysis of how the weights of losses $L_{Reg}$ and $L_{Gen}$ influence the continual adaptation process would be beneficial.**
>
> Response:
> We conduct a hyperparameter study on the loss weights $\lambda_1$ and $\lambda_2$ (initially set as 1,1), which control the two auxiliary losses---$L_{Reg}$ and $L_{Gen}$. The results are summarized in the table below:
>
> | L_Reg | L_Gen | Twitch | FB100 | Elliptic | OGB-Arxiv |
> | --- | --- | --- | --- | --- | --- |
> | 1 | 2 | 55.62±0.23 | 52.33±0.87 | 56.21±0.41 | 45.18±0.24 |
> | 1 | 3 | 55.56±0.15 | 52.48±0.11 | 56.17±0.38 | 45.12±0.23 |
> | 2 | 1 | 55.59±0.19 | 52.43±0.64 | 56.23±0.30 | 45.14±0.22 |
> | 3 | 1 | 55.55±0.18 | 52.69±0.38 | 56.27±0.23 | 45.15±0.26 |
>
> Across all datasets and configurations, the performance remains relatively stable, with only minor variations (generally within ±0.1%–0.3%). This indicates that GCAL is robust to moderate fluctuations in the weighting of these auxiliary objectives.
>
> > **Q1. Why are the AF results of some baseline N/A in Table 2?**
>
> Thank you for your inquiry. These results are marked as N/A because certain methods like EERM and GTrans operate by training a new set of parameters for each graph independently, without updating a model across multiple domains. Consequently, there is no shared memory or parameter set across tasks, thus eliminating the concept of "forgetting" as typically measured in continual learning scenarios. Thus, there is no way to measure their average forgetting.

---

### Official Review · Reviewer_YBfY · 2025-03-13

**Overall Recommendation:** 3

**Summary:**

This paper introduces **Graph Continual Adaptive Learning (GCAL)**, a novel framework for continual domain adaptation in graph models, specifically addressing challenges in adapting to multiple out-of-distribution (OOD) graph shifts. The method employs a bilevel optimization strategy with two phases: (1) **Adaptation**, using information maximization for self-supervised adaptation while mitigating catastrophic forgetting via memory replay, and (2) **Memory Generation**, utilizing a variational memory graph generation module based on an information bottleneck lower bound. The paper demonstrates through extensive experiments that GCAL outperforms existing state-of-the-art methods in continual graph adaptation.

## update after rebuttal

Thanks the authors for their rebuttal, I keep my score unchanged.

**Claims And Evidence:**

Overall, the submission provides substantial empirical and theoretical support for its claims. However, there are a few areas where the claims could be better substantiated or require additional clarification.

Some weakly supported claims:
1. **GCAL is efficient for continual adaptation in large-scale graphs.**
   - The paper does **not** provide computational complexity analysis or runtime benchmarks comparing GCAL to existing approaches.
   - Since bilevel optimization and variational memory graph generation introduce **additional computational overhead**, the authors should provide evidence on **training/inference time**, particularly for large-scale graphs.
   - Suggested improvement: Include runtime comparisons against CoTTA, EERM, or GTrans to demonstrate computational feasibility.

2. **Variational memory graph generation leads to significantly better knowledge retention.**
   - While the **ablation study** confirms that removing memory generation reduces performance, it is unclear **how much variational memory generation improves over simpler alternatives** (e.g., naive replay of stored subgraphs).
   - Suggested improvement: Compare GCAL’s memory generation against a **simpler heuristic-based memory selection** to isolate the exact benefits of the variational approach.

3. **GCAL can generalize well across different types of OOD shifts.**
   - The paper only evaluates **two types of shifts** (regional and temporal), which, while useful, do not fully represent all real-world graph distribution shifts (e.g., feature shifts, structural perturbations).
   - Suggested improvement: Add experiments on **synthetically perturbed graphs** to test robustness against **node feature corruption, edge rewiring, or adversarial attacks**.


Most claims in the paper are well-supported with empirical results and theoretical justification. Addressing the above gaps would make the claims more robust and convincing.

**Essential References Not Discussed:**

N/A

**Experimental Designs Or Analyses:**

I carefully read their experimental analysis and I think their analysis is reasonable

**Methods And Evaluation Criteria:**

The Methods and Evaluation Criteria Largely Make Sense. Some Areas Need More Justification or Alternative Evaluations

1. **Lack of Large-Scale Graph Evaluation**
   - The datasets used (Twitch, Facebook-100, OGB-Arxiv, Elliptic) have **relatively moderate-scale graphs** (up to hundreds of thousands of edges).
   - **Real-world continual graph adaptation problems (e.g., social media networks, citation networks, e-commerce graphs) often involve millions of nodes and edges.**
   - **Suggestion:** Evaluate GCAL on **larger-scale dynamic graphs** such as:
     - **Reddit (social interactions, time-evolving)**
     - **MAG-Scholar (large citation network)**
     - **Amazon/Alibaba (e-commerce graphs, evolving product-user interactions)**

2. **Computational Efficiency Not Evaluated**
   - **Bilevel optimization and variational memory graph generation** add complexity.
   - The paper does **not** provide a **runtime analysis** or **memory usage comparison** against baselines.
   - **Suggestion:** Report **training time, inference time, and memory footprint** compared to simpler adaptation methods (e.g., CoTTA, GTrans).

3. **No Evaluation on Feature or Structural Distribution Shifts**
   - The datasets primarily evaluate **temporal and regional shifts**, but in real-world applications, **feature shifts** (e.g., node attribute changes) and **structural shifts** (e.g., edge rewiring, node insertion/deletion) are common.
   -  **Suggestion:** Test GCAL on **synthetic or adversarial perturbations** to evaluate robustness under feature and structure shifts.
     - **Perturb node features (e.g., Gaussian noise, dropout).**
     - **Rewire graph structures (e.g., edge deletion/addition, graph sparsification).**

**Other Comments Or Suggestions:**

N/A

**Other Strengths And Weaknesses:**

N/A

**Questions For Authors:**

N/A

**Relation To Broader Scientific Literature:**

Related Prior Work like graph prompt learning [1-4] explores prompting techniques for cross-task generalization in GNNs. These methods suggest that prompt-based approaches can enable few-shot adaptation to new graph distributions. While GCAL does not explicitly use graph prompting, its memory replay approach serves a similar function—storing distilled graph representations for future adaptation. It would be more solid if they could include such a discussion in their related work section [2]. and give a summary of future work like exploring combining GCAL’s memory generation with graph prompting techniques to improve adaptability [1,3,4].
- [1] All in One: Multi-task Prompting for Graph Neural Networks. KDD 2023.
- [2] All in One and One for All: A Simple yet Effective Method towards Cross-domain Graph Pretraining. KDD2024.
- [3] Graph Prompt Learning: A Comprehensive Survey and Beyond. https://arxiv.org/abs/2410.01635
- [4] Does Graph Prompt Work? A Data Operation Perspective with Theoretical Analysis. https://arxiv.org/abs/2410.01635

**Theoretical Claims:**

I did not check the correctness of their theoretical proofs and I assume they are all correct.

---

> ### Author Rebuttal · Authors · 2025-03-31
>
> > **Computational Efficiency**
>
> Thank you for your valuable feedback. Our approach leverages a variational-based generation strategy, which is inherently designed to be efficient and scalable. This strategy allows for effective memory graph generation without significantly increasing the computational complexity. Thus, as the **time complexity analysis** in the Response to Reviewer `izUY` shows, the computation cost of the memory graph generator and subsequent losses is significantly less than the propagation of the GNN backbone, indicating the efficiency of our method.
>
> Following your suggestion, we have conducted extensive experiments to compare the running times of GCAL with CoTTA and GTrans training across four datasets on four 4090 GPUs. The results of these experiments are summarized in the following table.
>
> | Time / Seconds | CoTTA | GTrans | GCAL |
> | --- | --- | --- | --- |
> | Twitch | 41.6537 | 34.2958 | 26.842674 |
> | FB100 | 156.3651 | OOM | 41.446378 |
> | Elliptic | 26.5828 | 32.1564 | 22.967516 |
> | OGB-Arxiv | 44.4698 | 38.3925 | 40.146278 |
>
> These results demonstrate that GCAL not only operates within a competitive time frame but also significantly outperforms the baseline methods on multiple datasets. This evidence supports GCAL's effectiveness in this task.
>
> Regarding the scale of the datasets, the choice of these four datasets is aligned with the **established benchmarks** in studies related to out-of-distribution graph generalization [1,2]. We would like to clarify that the numbers of nodes and edges, as indicated in Table 1, represent the range within each dataset, where each dataset comprises multiple graphs. The overall number of nodes and edges across these datasets is indeed **substantial**. For example, Elliptic comprises a total of $189,033$ nodes and $217,223$ edges, and Facebook-100 consists of $157,921$ nodes and $13,197,698$ edges. Thus, these datasets meet the large-scale criteria to some extent. We hope this response adequately addresses your concerns.
>
> [1] Wu Q, et al. Handling Distribution Shifts on Graphs: An Invariance Perspective, ICLR 2022
>
> [2] Jin W, et al. Empowering Graph Representation Learning with Test-Time Graph Transformation, ICLR 2023
>
> > **Comparison to heuristic-based memory selection**
>
> Our approach in the unsupervised and out-of-distribution setting **diverges** from traditional continual learning frameworks, which typically assume that labels are provided and focus on adapting models to incremental classes or tasks. Consequently, many memory selection methods that rely on label information in standard continual learning settings are not applicable to our context.
>
> Following your suggestion, we have used a widely recognized heuristic-based memory selection method, K-Center[3], into our framework for comparative evaluation. This method selects K-representative data points as centers without the need for labels. The comparative results are in the table below:
>
> | Method | Twitch | FB100 | Elliptic | OGB-Arxiv |
> |--------|--------|-------|----------|-----------|
> | K-Center | 54.74±0.25 | 51.88±0.28 | 54.37±0.26 | 43.18±0.38 |
> | GCAL | 55.65±0.09 | 52.72±0.36 | 56.57±0.14 | 45.22±0.17 |
>
> The results show that our variational memory graph generation method overall outperforms K-Center in our framework, demonstrating its effectiveness in learning meaningful graph memory.
>
> [3] Nguyen, Cuong V., et al. "Variational Continual Learning."  ICLR 2018.
>
> > **Evaluation of Synthetic Feature or Structural Distribution Shifts**
>
> Thank you for your thoughtful feedback. The datasets used in our experiments inherently **involve both feature and structural shifts** across domains. For instance, social networks from different universities Facebook-100 exhibit variations in node attributes (e.g., user demographics) and structural patterns (e.g., friendship density). Citation networks OGB-Arxiv evolve over time, with node features (e.g., paper topics) and citation structures changing as research trends progress. These **real-world shifts** align with the challenges GCAL aims to address, validating its ability to adapt to combined feature and structural distribution shifts.
>
> Synthetic perturbations (e.g., Gaussian noise, edge rewiring) usually are more valuable for testing the robustness of graph neural networks, like against adversarial attacks. Our focus is on addressing practical, real-world OOD challenges within the continual adaptation framework using established benchmarks.
>
> Thank you again for your constructive feedback. Due to the time limitation, we will explore this direction in future research.
>
>
>
>
> > **Additional References**
>
> Thank you for your valuable feedback. As suggested, we will include a discussion of graph prompting techniques in Related Works, incorporating your highlighted references to provide a more comprehensive literature review.

---

### Official Review · Reviewer_izUY · 2025-03-13

**Overall Recommendation:** 4

**Summary:**

This paper proposes GCAL, a continual graph domain adaptation framework that mitigates catastrophic forgetting through bilevel optimization, integrating information maximization for adaptation and variational memory graph generation for knowledge replay. The approach is theoretically grounded in information bottleneck theory. Extensive experiments on multiple datasets demonstrate that GCAL outperforms baselines.

**Claims And Evidence:**

Overall, the paper presents well-supported claims with theoretical support and superior experimental performance.

**Essential References Not Discussed:**

N/A

**Experimental Designs Or Analyses:**

The experimental designs appear sound. But a time complexity analysis would be helpful.

**Methods And Evaluation Criteria:**

The proposed GCAL framework and evaluation criteria are well-aligned with the problem of continual graph domain adaptation

**Other Comments Or Suggestions:**

N/A

**Other Strengths And Weaknesses:**

Strengths

1. The studied problem is new and practical.

2. The proposed model is supported by a sound theoretical foundation based on information bottleneck theory.

2. The paper introduces a novel variational memory graph generation method for graph continual domain adaptation.


Weaknesses

1. The memory replay framework is commonly used in continual learning research. This paper does not introduce a fundamentally new framework in this regard.

2. This paper uses an adaptation learning objective to condense the graphs into memory graphs. The adaptation loss may not adequately capture the necessary structural or semantic information required for high-quality condensation.

3. The multiple learning objectives add to the complexity of this method. The time complexity of this method should be provided.

4. Some of the latest literature for graph domain adaptation is not included.

**Questions For Authors:**

See the weaknesses.

**Relation To Broader Scientific Literature:**

GCAL contributes a novel framework for continual graph domain adaptation, related to the research topics of continual graph learning, domain adaptation, and graph condensation.

**Theoretical Claims:**

The paper presents a theoretical lower bound based on the information bottleneck theory to support memory graph generation.

---

> ### Author Rebuttal · Authors · 2025-03-27
>
> > **W1: The memory replay framework is commonly used in continual learning research. This paper does not introduce a fundamentally new framework in this regard.**
>
> We acknowledge that memory replay is indeed a well-known approach to continual learning. We would like to clarify that **our novelty specifically lies in how the replay mechanism is integrated within the context of evolving out-of-distribution graphs**. Unlike existing methods that rely on stored raw data or labeled samples for replay, GCAL introduces a variational information bottleneck-based graph generator that creates synthetic memory graphs. This component is theoretically grounded (Eq. 3–5 in Sec. 3.2.1) and uniquely capable of generating compact, informative, and generalizable memory graphs in an unsupervised manner—a critical advancement where labeled data is not available.
>
> Furthermore, existing continual learning replay techniques primarily target Euclidean data formats. Our design includes graph condensation, Gumbel-softmax-based differentiable edge sampling, and gradient-matching-based memory optimization, which are specifically tailored for graph topologies, accounting for both structural and feature-level preservation (Sec. 3.2.2–3.2.5).
>
> > **W2: This paper uses an adaptation learning objective to condense the graphs into memory graphs. The adaptation loss may not adequately capture the necessary structural or semantic information required for high-quality condensation.**
>
> We appreciate this valuable observation regarding the sufficiency of the adaptation loss for graph condensation. We would like to clarify that the adaptation loss is only one component of a multi-objective memory graph learning strategy in GCAL. In particular, **the quality and informativeness of the memory graphs are ensured combination of three specialized losses with theoretical grounding**. We leverage the information bottleneck principle to derive three loss functions, each designed to explicitly preserve structural and semantic characteristics of the original graphs.
>
> As detailed in Sections 3.2.3–3.2.5 of the paper, the generation of memory graphs is not solely guided by the adaptation loss. The condensation loss $L_{MGL}$ leverages gradient matching to ensure the generated memory graphs induce similar optimization trajectories (gradients) as the original graphs. Rooted in variational inference (Eq. 12–13), this KL divergence-based term controls the latent distribution and promotes stability and informativeness in node and edge generation, avoiding overfitting to spurious patterns. The generation loss (Eq. 14) minimizes the distributional discrepancy between the memory graph and the original graph in the model’s latent space.
>
>
> > **W3: The multiple learning objectives add to the complexity of this method. The time complexity of this method should be provided.**
>
> For time complexity, we use GCNs as the backbone. The propagation cost is $O(L N_t d h + L N_t h^2)$, where $L$ is the number of layers, $N_t$ is the number of nodes, $d$ is the average degree, and $h$ is the hidden dimension. For the memory graph generation part, the TopKselector involving a sorting costs $O(N_t\log K+  N_t h)$, the construction and reparameterization in Eq.7,8 costs $O(K h)$ and $K^2 h$. The loss computations in Eq. 10, 13, and 14 cost $O(L h^2)$, $O(Kh +K^2)$, and $O(N_th + Kh)$. Because $K<<N_t$, the computation cost of the memory graph generator and subsequent losses is **significantly less** than the propagation of the GNN backbone, indicating the efficiency of our method.
>
>
> > **W4: Some of the latest literature for graph domain adaptation is not included.**
>
> Thank you for the valuable observation. We will update our related work section to incorporate the latest literature, ensuring a more comprehensive overview of graph domain adaptation methods.

---

### Official Review · Reviewer_GL4u · 2025-03-16

**Overall Recommendation:** 3

**Summary:**

This paper proposes Graph Adaptive Continual Learning (GCAL), extending the graph domain adaptation from single-step adaptation to continuous adaptation over a sequence of multiple domains. The proposed GCAL adopts a bi-level optimization strategy and consists of two phases. The adapt phase fine-tunes the given graph model on new graph domains based on information maximization, and the generate memory phase condenses the original graphs into memories, which will be used in future adapt phases to avoid forgetting.

The proposed method is evaluated on 4 public datasets.

**Claims And Evidence:**

Yes

**Essential References Not Discussed:**

No.

**Experimental Designs Or Analyses:**

For task construction, it is unclear how the adopted datasets are constructed into different tasks with different distributions.

In Table 2, actually most methods have a similar performance with Test, which is the lower bound, this is weird.

**Methods And Evaluation Criteria:**

Yes

**Other Comments Or Suggestions:**

See above.

**Other Strengths And Weaknesses:**

Strengths:

The proposed method resolves a weakness of the existing domain adaptation works, which is the limitation to single-step adaptation. The targeted continual multi-domain adaptation is more practical in real-world applications.

The proposed generate memory phase is supported by theoretical analysis.

The proposed method outperforms the baselines on all datasets.

Weakness:

For task construction, it is unclear how the adopted datasets are constructed into different tasks with different distributions.

In Table 2, most methods have a similar performance with Test, which is the lower bound; this is weird.

**Questions For Authors:**

How are the datasets constructed into different domains, and how to ensure that the different domains have different distribution?

What does #Nodes and #Egdes mean in Table 1? Why is it a range?

**Relation To Broader Scientific Literature:**

Continual graph domain adaptation learning is broadly related to different application scenarios involving evolving graph data.

**Theoretical Claims:**

Yes

---

> ### Author Rebuttal · Authors · 2025-03-27
>
> > **W1: For task construction, it is unclear how the adopted datasets are constructed into different tasks with different distributions.**
> **Q1: How are the datasets constructed into different domains, and how to ensure that the different domains have different distribution?**
>
> We appreciate the reviewer’s valuable feedback on our task construction. In our study, we selected datasets from established benchmarks widely used in research on graph out-of-distribution generalization [1,2]. **Each dataset comprises multiple real-world graphs, with each graph considered an independent domain**. We organize these graphs sequentially, based on regional differences and temporal shifts, to construct a continual adaptation setting.
>
> Appendix B.1 and B.3 provide a comprehensive explanation of dataset construction and partitioning. Specifically, Facebook-100 dataset consists of 100 separate Facebook friendship networks, each representing a distinct American university. Twitch-Explicit dataset includes seven networks on Twitch, sourced from different regions, such as France, Germany, and Russia. In these two datasets, nodes represent users, and edges denote friendships. OGB-Arxiv dataset encompasses 169,343 Arxiv CS papers across 40 subject areas to construct their citation networks, with the graphs divided by publication years. Elliptic dataset features 49 sequential graph snapshots of Bitcoin transaction networks, which are evenly spaced with an interval of about two weeks, where nodes represent individual transactions and edges indicate the flow of payments. For each dataset, we selected multiple former source domains for pre-training and selected the remaining graphs as sequential target domains to facilitate continual adaptation under varying distributions.
>
> These datasets are widely recognized for exhibiting distribution discrepancies between their graphs in the literature. From Figure 1, we also have **empirical evidence** supporting that the graphs indeed have different distributions.
>
> The links for downloading the datasets and the code for dataset processing are provided in the anonymous link in the Abstract to ensure the reproducibility of our study.
>
> [1] Wu Q, et al. Handling Distribution Shifts on Graphs: An Invariance Perspective, ICLR 2022
> [2] Jin W, et al. Empowering Graph Representation Learning with Test-Time Graph Transformation, ICLR 2023
>
>
> > **W2: In Table 2, most methods have a similar performance with Test, which is the lower bound; this is weird.**
>
> We appreciate the reviewer’s insightful comment. To clarify this point, the metric we used, "Average Performance" (AP) and "Average Forgetting" (AF), assesses model performance across all previously encountered domains. The phenomenon that many baseline methods exhibit similar performance to the "Test" occurs because common adaptation methods do not fully address the continual adaptation problem. They tend to experience substantial forgetting when adapting to new tasks. As these models continually update their parameters, they easily forget previously learned tasks, significantly reducing overall performance. Conversely, the "Test" method does not update or fine-tune its model parameters. This lack of adaptation makes stable, though still low, performance across all datasets.
>
> Our proposed method, GCAL, distinguishes itself by specifically addressing these challenges through memory replay and variational memory graph generation, which effectively retain and reuse previously learned information, enabling improved performance.
>
> This phenomenon is **not unique** to our study and indeed exists in the broader field of continuous learning. For example, in a different continual learning setting on graphs—class-incremental graph learning[3], certain baselines perform similarly or even worse than the lower bound. This is because baseline methods often collapse quickly due to catastrophic forgetting when continuously learning new tasks.
>
> [3] Zhang, Xikun, Dongjin Song, and Dacheng Tao. "Cglb: Benchmark tasks for continual graph learning." Advances in Neural Information Processing Systems 35 (2022): 13006-13021.
>
> > **Q2: What does #Nodes and #Egdes mean in Table 1? Why is it a range?**
>
> Thanks for the valuable questions. \#Nodes represents the number of nodes, and \#Edges means the number of edges in each graph. Since each dataset contains multiple graphs, each graph exhibiting different distributions, the number of nodes and edges varies across these graphs. Therefore, we report the range (minimum and maximum values) of nodes and edges present within the graphs of each dataset. We will make it clear by introducing this explicitly in the caption of Table 1 in the final version.

---

### Decision · Program_Chairs · 2025-05-01

**Decision:**

Accept (poster)

**Comment:**

**Summary:** This paper proposes GCAL, a bilevel optimization framework for continual graph domain adaptation under evolving out-of-distribution (OOD) shifts. GCAL alternates between two phases: (1) an Adapt phase based on information maximization for updating model parameters across new domains while mitigating forgetting, and (2) a Generate Memory phase that compresses source graphs into compact memory graphs via variational generation guided by an information bottleneck objective. The method is extensively evaluated across four datasets exhibiting regional and temporal shifts, and shows consistent improvements over baselines in terms of adaptability and knowledge retention.

**Decision:** This paper received uniformly positive evaluations from all four reviewers, with two clear accept scores and two weak accepts. While some concerns were raised—such as the absence of runtime comparisons, limited novelty given existing continual learning techniques, etc.—the authors addressed these in detail. Given the overall positive reception, the clear technical contributions, and the comprehensive rebuttal, I recommend acceptance.